

**Network complexity of rubber plantations is lower than tropical forests for soil**
**bacteria but not fungi**
Guoyu Lan[a,b*], Chuan Yang [a,b] , Zhixiang Wu [a,b]
a. Rubber Research Institute, Chinese Academy of Tropical Agricultural Sciences,
Danzhou City, Hainan Province, 571737, P. R. China;
b. Danzhou Investigation & Experiment Station of Tropical Crops, Ministry of
Agriculture and Rural Affairs, Danzhou City, Hainan Province, 571737, P. R.
China
**Running title**: Forest conversion alters soil microbial networks
*Correspondence
Dr. Guoyu LAN
Tel: +86-898-23301800
Fax: +86-898-23300315
E-mail: langyrri@163.com





**Abstract:**
Soil microbial communities play a crucial role in ecosystem functioning. Past
research has examined the effects of forest conversion on soil microbial composition
and diversity, but it remains unknown how networks within these communities
respond to forest conversion such as when tropical rainforest are replaced with rubber
plantations. In this study, we used Illumina sequencing and metagenome
shotgun sequencing to analyze bacterial and fungal community network structure in a
large number of soil samples from tropical rainforest and rubber plantation sites in
Hainan Island, China. Our results showed only a few shared network edges were
observed in both bacterial and fungal communities, which indicates that forest
conversion altered soil microbial network structure. We found a greater degree of
network structure and a larger number of network edges among bacterial networks in
samples from tropical rainforest compared to samples from rubber plantations. The
difference was especially pronounced during the rainy season and indicates that
rainforest bacterial networks were more complex than rubber plantation bacterial
networks. However, rubber plantations soil fungal networks showed more higher
links and higher network degree, suggesting that forest conversion does not reduce
fungal network complexity. We found that some groups of Acidobacteria were
keystone taxa in our tropical rainforest soils, while Actinobacteria were keystone taxa
in rubber plantation soils. In addition, seasonal change had a strong effect on network
degree, the complexity of soil bacterial and fungal network structure. In conclusion,
forest conversion changed soil pH and other soil properties, such as available



potassium (AK) and total nitrogen (TN), which resulted in changes in bacterial and
fungal network composition and structure.

**Keyword:** Tropical rainforest, Rubber plantations, Networks, Soil microbial
community, Forest conversion






## 1. Introduction


Tropical rainforest have the highest biodiversity of any ecosystem and harbor more
than 60% of all known plant and animal species (Dirzo and Raven, 2003). However,
over the past several decades, logging, mining, slash and burn agriculture have caused
widespread deforestation and forest degradation. Of these,the conversion of forest to
agriculture has caused the most forest loss (Li et al., 2007).
Hainan is home to a large area of tropical rainforest rich in biodiversity. It is a part
of the Indian-Malay rainforest system and the northern edge of the world's rainforest
distribution. However, rubber plantations now account for almost a quarter of the total
extent of vegetated areas on Hainan Island (Lan et al., 2020a).
The soil microbiome is highly diverse and comprises up to one quarter of Earth's
diversity (Wagg et al., 2019). Soil microbes play a critical role in the maintenance of
soil quality and function, and they represent the majority of biodiversity in terrestrial
ecosystems (Philippot et al., 2013). A number of studies have investigated the impact
of the conversion of tropical forests to rubber plantations on soil microbial
composition and diversity (Schneider et al., 2015; Kerfahi et al., 2016, Lan et al.,
2017a; 2017b; 2017c; Lan et al., 2020a; 2020b; 2020c). Studies conducted in
Indonesia (Schneider et al. 2015), Malaysia (Kerfahi et al. 2016) and South China
(Lan et al. 2017a; Lan et al. 2017b; Lan et al. 2017c) have found significant
differences between rubber plantations and tropical forests, specifically that the
diversity of soil bacteria was higher in rubber plantations than in rainforest. Compared
to primary forests, agricultural systems tend to have higher bacterial richness but





lower fungal richness (Lan et al., 2017a; Cai et al., 2018; Tripathi et al., 2012; Kerfahi
et al., 2016). However, there are few studies on the effects of forest conversion on soil
microbial network structure.
Network analysis is an increasingly popular tool for investigating microbial
community structure, as it integrates multiple types of information and may represent
systems-level behavior (Röttjers and Faust, 2018).  The soil microbial network is
viewed as a critical indicator of soil health and quality (Kuperman et al., 2014).
Network analysis of taxon co-occurrence patterns provides new insight into the
structure of complex microbial communities, insight that complements and expands
on the information provided by the more standard suite of analytical approaches
(Barberan et al., 2012). Previous work has shown that agricultural intensification can
reduce microbial network complexity (Banerjee et al., 2019). Logging alters soil
fungal network in tropical rainforests, i.e., a better-organized fungal community in the
select cut stands when compared with the primary stands (Chen et al., 2019). Soil
bacterial networks are less stable under drought than fungal networks (De Vries et al.,
2018). Soil networks become more connected as ecological restoration progresses
(Morriën et al., 2017). So far, very few studies have assessed the impact of forest
conversion on soil microbial networks and it is still unclear whether forest conversion
as well as seasonal change influences the structure and complexity of microbial
networks. Here we explored bacterial and fungal community network structure using
Illumina sequencing based on samples collected from tropical rainforest and rubber
plantations in Hainan Island, China. We aimed to test the hypothesis that (1) forest



conversion alters microbial networks by altering microbial community composition [3]
(Lan et al., 2020a) and that soil microbial activity is strongly influenced by plant
species (Galicia and García-Oliva 2004). (2) Soil microbial network structure in
rainforest sites is more complex and stable than in rubber plantations because natural
systems were more connected than artificial systems (Morriën et al., 2017). By
evaluating these hypotheses, we want to clarify the drivers and mechanisms that link
forest conversion to differences in soil microbial network structure . This study will
provide critical information for understanding and managing microbial communities
in tropical forests of China and elsewhere.
**2. Methods**
**2.1 Study site**
This study was conducted on Hainan Island (18°10′–20°10′N and 108°37′–111°03′E),
south China. The total area of Hainan Island is about 34,000 km$^2$ (Lopez et al., 2009).
Hainan Island is the largest island within the Indo-Burma Biodiversity Hotspot in
tropical Asia (Francisco-Ortega et al., 2010) and has a tropical monsoon climate.
Hainan Island has a warm and humid climate all year round, with an average annual
temperature of 22-26°C. The rainy season occurs from May to October, with a total
precipitation of about 1500 mm, accounting for 70-90% of the total annual
precipitation. Only 10-30% of the total annual precipitation falls within the dry
season, from November to April. Rainfall is abundant, ranging from 1,000 mm to
2,600 mm yearly, with an average annual precipitation of 1,639 mm. The central part
of Hainan Island is mountainous and contains old-growth tropical rainforests and





monsoon forests. Rubber plantations are found on the plateaus surrounding the
central mountainous zone.
**2.2 Soil sampling**
We selected five rainforests as our study sites: Bangwang mountain, Diaoluo
mountain, Wuzhi mountain, Yinge mountain and Jianfeng mountain. Five rubber
plantations were selected in Haikou, Danzhou, Qiongzhong, Wanning and Ledong
(Figure S1). More information on the study sites is provided in Table S1. For each
site, thirteen soil samples were collected, thus there were a total of 130 samples
collected between the rubber plantations and tropical rainforest per sampling interval.
Soil sampling was performed twice in 2018, once in the rainy season (July) and once
in the dry season (January). Thus, there were a total of 260 soil samples (130 per
forest type). Soil samples were divided into two parts: one was used to analyze soil
water contents, soil pH, total nitrogen, total phosphorus (TP), total potassium (TK),
nitrate nitrogen (NN), ammonium nitrogen (AN), available phosphorus (AP),
potassium (AK). The other was used for DNA extraction. Soil properties were
analyzed following the methods described in by Lan et al. (2020b). Soil properties of
the rubber plantation and rainforest sites are shown in Table S2.
**2.3 DNA extraction and PCR amplification**
Microbial DNA was extracted from 0.5 g of soil using the E.Z.N.A.® Soil DNA Kit
(Omega Bio-tek, Norcross, GA, U.S.) following the manufacturer's protocol. The
fungal ITS1 hypervariable region was amplified using the PCR primers ITS1F
(5'-CTTGGTCATTTAGAGGAAGTAA-3') and ITS2R

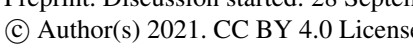



(5'-GCTGCGTTCTTCATCGATGC-3') (Adams et al., 2013). For bacteria and
archaea, the V4 hypervariable region of the bacterial 16S rRNA gene was amplified
using the PCR primers 515FmodF (5'-GTGYCAGCMGCCGCGGTAA-3') and
806RmodR (5'-GGACTACNVGGGTWTCTAAT-3') (Walters et al., 2016; Sampson
et al., 2016). The PCR reactions were conducted using the following approach: an
initial 3 min denaturation at 95°C; followed by 27 cycles of 30s at 95°C, 30s of
annealing at 55°C, and 45s of elongation at 72°C; and a 10 min final extension at
72°C.
**2.4 Illumina MiSeq sequencing**
Amplicons were extracted from 2% agarose gels, purified using the AxyPrep DNA
Gel Extraction Kit (Axygen Biosciences, Union City, CA, U.S.) and quantified using
a QuantiFluor™ -ST Fluorometer (Promega, U.S.). Purified amplicons were pooled in
an equimolar solution and then sequenced (paired-end, $2 \times 250$ bp) on an Illumina
MiSeq platform according to standard protocols.
Metagenomic shotgun sequencing libraries were prepared and then sequenced by
Majorbio, Inc. (Shanghai, China) using the Illumina HiSeq 2000 platform. The NR
gene catalog was aligned against the Kyoto Encyclopedia of Genes and Genomes
(KEGG) database using BLAST (Version 2.2.28+) and then functionally annotated
using KOBAAS 2.0 according to previously described methods (Qin et al., 2010)
**2.5 Bioinformatics and data analysis**
Raw fastq files were demultiplexed and quality-filtered using QIIME (Caporaso et al.,
2010) (version 1.17). Operational Taxonomic Units (OTUs) were clustered with a



97% similarity cut-off using UPARSE (Edgar, 2013), and chimeric sequences were
identified and removed using UCHIME. Using the RDP Classifier, the phylogenetic
affiliation of each 16S rRNA gene and ITS gene sequence was determined using a
confidence threshold of 70% with the SILVA 16S rRNA database and UNITE
database, respectively (Amato et al., 2013). The relative abundance was determined
for each taxon (Good, 1953), and the Shannon and Simpson diversity indices were
calculated based on re-sampled sequence data using MOTHUR (Schloss et al.,
2009). For each site, the relative abundance of different taxa (Good, 1953) and the
Shannon diversity index were calculated based on re-sampled sequence data using
MOTHUR (Schloss et al., 2009).
**2.6 Statistical analysis**
For the co-occurrence network analyses, we only focus on the top 300 most abundant
OTUs of the two forest types. The networks of each habitat during each sampling
period (tropical rainforest and rubber plantations in dry season and rainy season) were
constructed with 65 samples each. Interactions consisted of Spearman's rank
correlations and co-occurrence networks were constructed using only significant
correlations of $\rho > 0.6$ ($P < 0.01$) (Barberan et al. 2012), because this cutoff includes
a range of interactions strengths (De Vries et al., 2018). The networks were then
visualized in R using the *igraph* package. To reveal the distribution pattern of
correlation coefficients, the frequency of the coefficients of $\rho > 0.3$ ($P < 0.01$) were
plotted. The Network Analyzer tool in Cytoscape (version 3.4.0) was used to calculate
network topology parameters including number of nodes, edges, degree, betweenness,



closeness. In order to evaluate the network differences between tropical rain forest
and rubber forest sites, Venn diagrams were plotted to reveal the number of shared
edges and unique edges which were calculated using *igraph*. Keystone OTUs were
selected on the basis of high network degree, high closeness centrality, and low
betweenness centrality as defined by Berry and Widder (2014). To evaluate the
proportional influence of each phylum on bacterial and fungal network structure, node
degrees of each phylum were calculated and bar plots were created. Correlation
coefficients between species and functions were calculated based on metagenomics
data. Here we used the top 50 most abundant species and top 50 KEGG functions
(pathway level 3). Then the species and function correlation network was constructed
on the Major bio cloud platform (https://cloud.majorbio.com/). To reveal the
relationship between microbial taxon and environment variables, two-way correlation
networks were also constructed on the Major bio cloud platform. The topological role
of each node in a network was assessed by the $Z_i$ and $P_i$ values, where $Z_i$ represents
the nodes connectivity within a module, and $P_i$ measures the degree of a node
connected with other modules (Roger and Amaral, 2005). All species can be divided
into four groups according to the simplified criteria (Olesen et al., 2007), namely
peripherals ($Z_i < 2.5$ and $P_i < 0.62$), connectors ($P_i > 0.62$), module hubs ($Z_i > 2.5$)
and network hubs ($Z_i > 2.5$ and $P_i > 0.62$). The $Z_i$ and $P_i$ values were calculated using
GIANT package of Cytoscape. The $Z_i$-$P_i$ plot was created with *ggplot2* in R.
**3. Results**
**3.1 Bacterial and fungal networks**

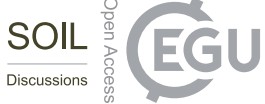

Our results showed most nodes of bacterial networks (Figure 1) and fungal networks
(Figure 2) varied with forest type in both the dry season and rainy season. For
bacterial networks, there were 2559 and 2501 edges in tropical rainforest and rubber
plantation in dry season respectively, but these two networks only shared 262 edges,
accounting only about 10% the total edges (Figure 3A-B). Similarly, these networks
only shared 519 edges during the rainy season. For fungal networks, there were only 4
and 5 shared edges (i.e., no more than 5% of the total edges) in dry season and rainy
season, respectively (Figure 3C-D).

The number of edges of bacterial and fungal networks were almost equivalent

during the dry season. However, in rainy season, there were more edges in the
bacterial network in tropical rainforest than in the rubber plantation (Table 1 & Figure
3B). For the network structure of the fungal community, more edges were observed in
rubber plantations in rainy season (Table 1 & Figure 3D). Similarly, there were no
significant differences in both bacterial and fungal network degree between tropical
rainforest sites and rubber plantations in the dry season (Figure 4A, C). In the rainy
season, rainforest sites had higher bacterial network degree, while rubber plantations
showed higher fungal network degree (Figure 4B, D). For bacterial networks, more
nodes (OTUs) with high degree (rubber plantation had 2 nodes with degree greater
than 75, rainforest had 8 such nodes) of rainforest were observed during the rainy
season (Figure S2B). For fungal networks, 15 nodes of higher degree (degree greater
than 25) were observed in rubber plantations, however, only 7 such nodes existed for
rainforest sites (Figure S2D). These results indicate rubber plantation fungal network



structure was more complex than tropical rainforest during the rainy season, but that
the reverse was true for bacteria.
When considering the ratio of positive to negative correlation coefficients, more
correlations (greater than 0.3, $P < 0.05$) were calculated, and the results showed that
the negative correlations between bacterial and fungal OTUs of rubber plantations
were consistently stronger than for tropical rainforest in both dry season and rainy
season (Figure 5).
For both the bacterial and fungal communities, neither tropical rainforest nor
rubber plantation networks possessed module hubs and network hubs (Figure S3-S4).
For bacterial network, the majority of nodes in both the rubber plantation and tropical
rainforest networks were connectors (Figure S3). However, for fungal networks, the
majority of the nodes in both rubber plantation and tropical rainforest networks were
peripherals and connectors (Figure S4). The ratio of peripherals and connectors of
these two forest types was not different indicating the network structures of rainforest
and rubber plantation were very similar as visualized in Figure 1 and Figure 2.
However, the bacterial networks had more connectors than fungal networks, which
suggests bacterial networks may contain more generalists than fungal networks do.
This indicated bacterial network were more complex than fungal network, which can
further confirmed by visualization of the network (Figure 1 and Figure 2).
For bacterial groups, members of the phyla Acidobateria, Planctomycetes and
Verrucomicrobia showed higher degree in the tropical rainforest sites than in rubber
plantations, suggesting that these taxa are strongly associated with the other members





of the community in tropical rainforest (Figure S5A). Members of the phyla
Actinobacteria showed higher degree in rubber plantations. Seasonal change also had
effects on network degree for soil bacterial networks. For instance, Chloroflex had
higher degree in rubber plantation in the dry season, but the opposite is true in the
rainy season. For fungal networks, members of Basidiomycota showed higher degree
in tropical rainforest sites duringin the dry season, however, Ascomycota showed
higher degree in rubber plantations (Figure S5C-D) during the rainy season.
We used total degree of each phylum to reveal the influence of each phylum on
network structure (Figure 6). For bacteria, Proteobacteria, Actinobacteria and
Acidobacteria had a large influence on network structure (Figure 6). Acidobacteria
and Planctomycetes contributed more to rainforest networks than rubber plantation
networks. However, Actinobacteria and Chloroflex showed the opposite. For fungi,
Ascomycota and Basidiomycota had large influence of network structure. Both
Ascomycota and Basidiomycota had stronger influence on rainforest networks than
rubber plantation networks. The influence of Ascomycota was stronger during the
rainy season than in the dry season, indicating seasonal change also had impact on
fungal community networks.
**3.2 Keystone taxa**
Keystone OTUs of the bacterial and fungal communities were selected on the basis of
high degree, high closeness centrality, and low betweenness centrality. The results
showed that forest conversion altered the keystone taxa of bacteria and fungi. The
keystone taxa of bacteria were very different between rubber plantations and tropical



rainforest sites in both the dry season and rainy season. For bacteria, there were more
keystone taxa in tropical rainforest sites than in rubber plantations in both the dry
season and rainy season indicating that the tropical rainforest networks had higher
complexity. We found that some groups of Acidobacteria are keystone taxa in tropical
rainforest sites but disappeared after forest conversion. There were more
Actinobacteria bacteria in rubber plantations than in tropical rainforest sites (Table
S3).
For fungi, more keystone taxa were observed in rubber plantations than in tropical
rainforest sites during both the dry season and rainy season, indicating the rubber
plantation networks were more complex. Most keystone taxa belong to Ascomycota
suggesting member of this group are very import for network structure. In addition to
forest conversion, seasonal changes also affect the keystone taxa of the fungal
community network. There were more Basidiomycota OTUs in the dry season, but
more Ascomycota in rainy season (Table S4).
**3.3 Two-ways correlation networks**
Two-way network analysis of the 50 most abundant species (metageomic data, the 50
most abundant species all belong to bacteria groups) and the 50 most abundant KEGG
functions revealed that soil microbial community structure in at rainforests sites was
more complex than rubber plantations. (Figure 7). Both rubber plantations and
rainforest networks were more complex in the rainy season than in dry season. We
also found that metabolism was the most important function in soil microbial network.
Surprisingly, species of Actinobacteria negatively correlated with other species and





function in rubber plantations (Figure 7).
Two-ways correlation network analysis revealed the interaction between microbial
composition and environmental variables. This analysis includes different
environmental factors as nodes in the network, and the number of connections these
nodes have indicates the number of OTUs that are impacted by that environmental
factor (Figure 8). For bacteria, elevation had the highest network degree at 106, and
was followed by AK (104), soil pH (86) and TK (9). In other words, elevations, AK,
soil pH are all drivers of bacterial community composition. Soil pH negatively
correlated with most bacterial Acidobcteria OTUs. For fungi, elevation had the
highest network degree (61), followed by AK (51), longitude (15), and NN (11). AK
positively correlated with most OTUs of Basidiomycota. Relationship between OTU
abundance and soil pH revealed the soil pH negatively correlated with members of
Acidobacteria, but positively correlated with members of Chloroflexi and members of
Ascomycota (Figure 9). AK positively correlated with members of Planctomycetes
Verrucomicrobia and Basidiomycota, however negatively correlated with Chloroflexi
and Ascomycota.

**4. Discussion**
**4.1 Forest conversion reduces soil bacterial network complex**
Land-use changes increasingly threaten biodiversity, particularly in tropical forests
(Gibson et al., 2011). However, we still have little understanding of how soil
networks responsd to forest conversion, such when rainforests are converted to rubber



plantations. Our results showed that forest conversion had large effects on both soil
bacterial and fungal networks. More edges (Table 1) and higher degree (Figure 4) of
tropical rainforest bacterial networks were observed, especially during the rainy
season, which indicates that the rainforest bacterial network was more complex than
the rubber plantation network. This consistent with previous observations that soil
bacterial networks were more complex in natural systems than in crop soil (Karimi et
al., 2019). Further study showed that soil networks become more connected as nature
restoration progresses (Morriën et al., 2017). The observed decrease in network
complexity and cohesion supports the hypothesis that cropping may enhance the
isolation of bacterial taxa (Karimi et al., 2019), which results in lower connection of
the network. In addition, at the microscale, the structure of tilled soils is more
homogeneous, and the soil pores are less connected than in soils under without tillage
(Pagliai et al., 2004), such as rainforest soil. In nature, soil ecosystems are highly
heterogeneous since soil microbial biodiversity hot spots can form spatial and
temporally within soil aggregates (Bach et al., 2018). This spatial heterogeneity likely
plays an important role for the interactions among microbes and the mechanisms by
which more complex and diverse communities drive various nutrient cycling
processes on small spatial scales (Wagg et al., 2018).

A large number of studies employing microbial network analysis have enriched

our understanding of microbial co-occurrence patterns in various soil ecosystems,
however, very little is known of whether differences in the structure of microbial
networks have consequences for microbiome functioning (Wagg et al., 2018). Our





results demonstrated that more species related with metabolism in natural system than
in the agricultural system, especially in the rainy season. This is in line with a
previous study conducted in Sumatra, Indonesia, which found that the transformation
of forest to rubber results in a 10-16% decrease in community metabolism (Barnes et
al., 2014). Fewer interactions between microbial species (most of them are bacteria)
and functions in rubber plantations demonstrated that forest conversion reduced soil
bacterial network complex.
**4.2 Forest conversion does not reduce soil fungal network complexity**
Surprisingly, rainforest bacterial networks were characterized by fewer edges (Table 1)
and lower degree (Figure 4), which means that rubber plantation bacterial networks
were more complex than the native forest. Although, our results were consistent with
previous observations which found that fungal community networks were better
organized disturbed forest compared to primary forest (Chen et al., 2019). Banerjee et
al. (2019)'s observation showed that organic agricultural fields harbored much more
complex fungal networks with many more keystone taxa than conventional managed
fields. Forest conversion resulted in shifts in fungal composition from Basidiomycota
to Ascomycota (Figure S7), as seen in previous investigations (Lan et al., 2020a; Lan
et al., 2020b). Previous work showed that Basidiomycota species show higher drought
sensitivity than Ascomycota species (Taniguchi et al., 2018), this would result in a
shift in richness and abundance of Basidiomycota species (Figure S6). Many
Basidiomycota species are capable of long-distance dispersal (Egidi et al. 2019, Geml
et al., 2012), which may result in a decrease in fungal network. This possibly





explained why Ascomycota OTUs contribute more to the network structure than
Basidiomycota (Figure 6). Overall, reduction in abundance and richness of
Basidiomycota species led to an increase in fungal links in rubber plantations.
**4.3 Forest conversion enhanced the stability of soil network**
The positive to negative ratio of network links indicates the balance between
facilitative and inhibitive relationships within a network (Karimi et al., 2017).
Theoretical studies, for example, predict that ecological networks that consist of weak
interactions are more stable than those with strong interactions (Neutel et al., 2002,
Coyte et al., 2015), and that compartmentalization and presence of negative
interactions increase the stability of networks under disturbances (Coyte et al., 2015,
Rooney et al., 2006, Stouffer & Bascompte 2011). In our study, more negative
correlations were detected in rubber plantation, indicating the network structure of
rubber plantation soils was more stable than rainforest soils (De Vries et al., 2018).
**4.4 Driver of the network structure**
Forest conversion results in the loss of plant diversity, plant biomass and increasing
soil pH (Lan et al., 2017a, 2017b). Rubber plantations had a significantly higher pH,
which explains the relative decrease in the abundance of Acidobacteria (Lan et al.,
2017a). Our results demonstrate that keystone taxa of soil microbes change after
forest conversion (Table 1). We found that many OTUs of Acidobacteria fit our
criteria as keystone species for rainforest sites, which is consistent with previous
findings (Banerjee et al., 2018). Unexpectedly, OTU11388 and OTU11373, both
Acidobacteria, were observed in rainforest soils in both the dry and rainy seasons,



indicating Acidobacteria were very important for rainforest soil bacterial networks
(Figure 6 and table S3). Higher AK concentration resulted in a higher abundance and
more taxa of of Actinobacteria (Figure 9), which suggests that Actinobacteria
contributed more in rubber plantation than in rainforest (Figure 6). Indeed, forest
conversion reduced the abundance of Actinobacteria OTUs (Figure S7) Due to the
human disturbance in rubber plantations, the soil will inevitably be slightly polluted
with herbicides and domestic garbage. Previous study showed member of
Actinobacteria were observed in contaminated soil (Jiao et al., 2016).
Forest conversion also increases land use intensity (Brinkmann et al., 2019),
including the    application of fertilizer and herbicide. Herbicide application also
caused significant decreases in root colonization and spore biomass of arbuscular
mycorrhizal fungi in tropical agriculture (Zaller et al., 2014). Soil nutrient
concentration shows a decline around the roots of rubber plantations compared to
those from rainforests (Sahner et al., 2015). Our observation is no exception, for
instance, AK and TN concentration was significant lower in rubber plantation than in
samples from rainforest sites (Table S2). Higher concentration of AK reasonably
explained the higher contribution of Basidiomycota on the network structure (Figure
8B) due to AK positive association with Basidiomycota.
Spatiotemporal heterogeneity can be a major driver of the abundance and
distribution of keystone taxa in soil which is a highly heterogeneous and multifaceted
environment (Mills et al., 1993, Power et al., 1996; Mouquet et al., 2013). Seasonal
variability determines the structural and compositional properties of microbiomes in





an environment, and as such, a keystone species might be present only in a specific
season or time period (Banerjee et al., 2018). It was interesting that more bacterial
OTUs were identified as connectors during the rainy season than in the dry season.
Connectors have been characterized as generalists (Olesen et al., 2007), and
generalists drive covariation among communities in a network (Chen et al., 2019).
Previous observation demonstrated that some keystone taxa that were found in the dry
season disappeared during the rainy season (Lan et al., 2018) . Seasonal changes
possibly explained the keystone taxa was observed in rainy season but not in dry
season.
**5.   Summary**
Our knowledge about land-use impacts on soil ecosystems is mostly limited to
biodiversity and ecosystem functions, leaving uncertainty about how soil networks
change after forest conversion. This study is the most comprehensive report on
changes in network structure that occur when tropical rainforests are converted into
rubber forest. Our study showed that forest conversion altered both bacterial and
fungal soil networks, reduced bacterial network complexity and enhanced fungal
network complexity, especially during the rainy season. One possible reason maybe
that forest conversion changed soil pH and other soil properties, which altered
bacterial composition and subsequent network structure. Our study demonstrates the
impact of forest conversion for soil network structure, which has important
implications for ecosystem functions and health of soil ecosystems in tropical regions.






**Availability of data and material**

The raw reads were deposited into the NCBI Sequence Read Archive (SRA) database

(Accession Number: SRP108394, SRP278296, SRP278319).

**Code availability**

Not applicable

**Authors' contributions**

Guoyu Lan: Conceptualization, Methodology, Writing- Reviewing and Editing;

Chuan Yang and Zhixiang Wu: Investigation

**Competing interests**

The authors declared that they have no conflicts of interest to this study.

**Disclaimer**

Publisher's note: Copernicus Publications remains neutral with regard to jurisdictional

claims in published maps and institutional affiliations.

**Acknowledgements**

We thank Dr. Tim Treuer for his assistance with English language and grammatical

editing.

**Financial support**

This work was supported by Finance Science and Technology Project of Hainan

Province (ZDYF2019145); National Natural Science Foundation of China

(31770661); High level Talents Project of Hainan Natural Science Foundation

(320RC733); the Earmarked Fund for China Agriculture Research System



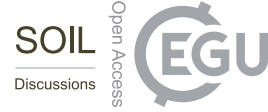

(CARS-33-ZP3)

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

**Figure legend**
**Figure 1** Soil bacterial network structure of rubber plantations and tropical rainforest
in dry and rainy seasons. Red lines indicate positive correlation between OTUs, and
green indicate negative correlation. Absolute value of correlation coefficient > 0.6, *P*
< 0.05
**Figure 2** Soil fungal network structure of rubber plantations and tropical rainforest in
dry and rainy season. Red lines indicate positive correlation between OTUs, and
green indicate negative correlation. Absolute value of correlation coefficient > 0.6, *P*
< 0.05
**Figure 3** Soil microbial (bacterial and fungal) network of tropical rainforest and
rubber plantations in dry season and rainy season.
**Figure 4** Network degree of soil bacterial and fungal community of rubber plantations
(blue) and tropical rainforest (red) in dry season and rainy season.
**Figure 5** Frequency distributions of correlations in bacterial (a: dry season, b: rainy
season) and fungal (c: dry season, d: rainy season) networks of rubber plantations and
tropical rainforest in the dry season and rainy season. (Absolute correlation coefficient





greater than 0.3, $P < 0.05$) Correlations in rainforest networks are red, correlations in
rubber plantation networks are blue.
**Figure 6** Proportional influence of different phylum on bacterial and fungal network
structure in both dry season and rainy. The influence was the number of degrees of
nodes belonging to a particular phylum. (a: bacteria in dry season, b: bacteria in rainy
season, c: fungal in dry season, d: fungal in rainy season).
**Figure 7** Network of the top 50 most abundant species (based on metagenomics data)
and top 50 most frequent KEGG functions (pathway level 3) of rubber plantations and
tropical rainforest sites in dry season and rainy season. (A: rubber in dry season; B:
rainforest in dry season; C: rubber in rainy season; D: rainforest in rainy season) The
size of the node indicates the species/function abundance. A red line indicates positive
correlation between species/functions, and green indicates negative correlation.
Absolute value of correlation coefficient > 0.6, p < 0.05
**Figure 8** Two ways correlation network of top 500 most abundant bacterial (A) and
fungal (B) OTUs and environmental factors. The size of the node indicates the OTU
abundance. A red line indicates positive correlation between species/functions, and
green indicates negative correlation. Absolute value of correlation coefficient > 0.5, p
< 0.05.
**Figure 9** Relationship between abundance of phylum (bacteria: A-E, I-M; fungi: F-H,
N-P) and soil properties (Soil pH: A-H; AK (available potassium) concentration: I-P)



**Table 1** Topological properties of soil microbial (bacterial and fungi) network
structure in rubber plantation and tropical rain forest in dry season and rainy season

| | Bacteria | | | | Fungi | | | |
|---|---|---|---|---|---|---|---|---|
| | Rubber | Rainforest | Rubber | Rainforest | Rubber | Rainforest | Rubber | Rainforest |
| | Dry | Dry | Rainy | Rainy | Dry | Dry | Rainy | Rainy |
| No. of nodes | 291 | 287 | 296 | 296 | 220 | 235 | 243 | 244 |
| No. of edges | 2448 | 2559 | 4248 | 5019 | 791 | 769 | 1250 | 905 |
| No. of positive edges | 2052 | 2508.00 | 3385 | 4901 | 760 | 764 | 1195 | 897 |
| No. of negative edges | 396 | 51 | 863 | 118.00 | 31 | 5 | 55 | 8 |
| Connectance | 0.06 | 0.06 | 0.09 | 0.11 | 0.02 | 0.02 | 0.03 | 0.02 |
| Average degree | 16.82 | 17.83 | 16.67 | 33.91 | 16.67 | 6.54 | 10.28 | 7.41 |
| Average betweenness | 100.61 | 93.52 | 31.23 | 42.57 | 208.90 | 185.03 | 170.99 | 266.31 |
| Average of shortest path length | 2.92 | 2.92 | 2.56 | 2.45 | 4.00 | 3.626 | 3.81 | 5.27 |
| Diameter | 6.00 | 7.00 | 7.00 | 6.00 | 10.00 | 12.00 | 9.00 | 15.00 |
| Cluster of coefficient | 0.46 | 0.451 | 0.51 | 0.54 | 0.59 | 0.457 | 0.49 | 0.51 |
| No of clusters | 11.00 | 14.00 | 5.00 | 5.00 | 94.00 | 75.00 | 62.00 | 67.00 |
| Degree centralization | 0.12 | 0.11 | 0.17 | 0.18 | 0.08 | 0.08 | 0.10 | 0.10 |
| Betweenness centralization. | 0.0066 | 0.0067 | 0.0053 | 0.0049 | 0.02 | 0.03 | 0.121 | 0.04 |
| Closeness centralization. | 0.35 | 0.35 | 0.40 | 0.41 | 0.31 | 0.351 | 0.291 | 0.26 |
| Neighborhood Connectivity | 21.12 | 21.79 | 34.69 | 40.68 | 10.45 | 9.28 | 13.78 | 9.44 |
| Topological coefficient | 0.26 | 0.26 | 0.26 | 0.27 | 0.33 | 0.36 | 0.37 | 0.50 |





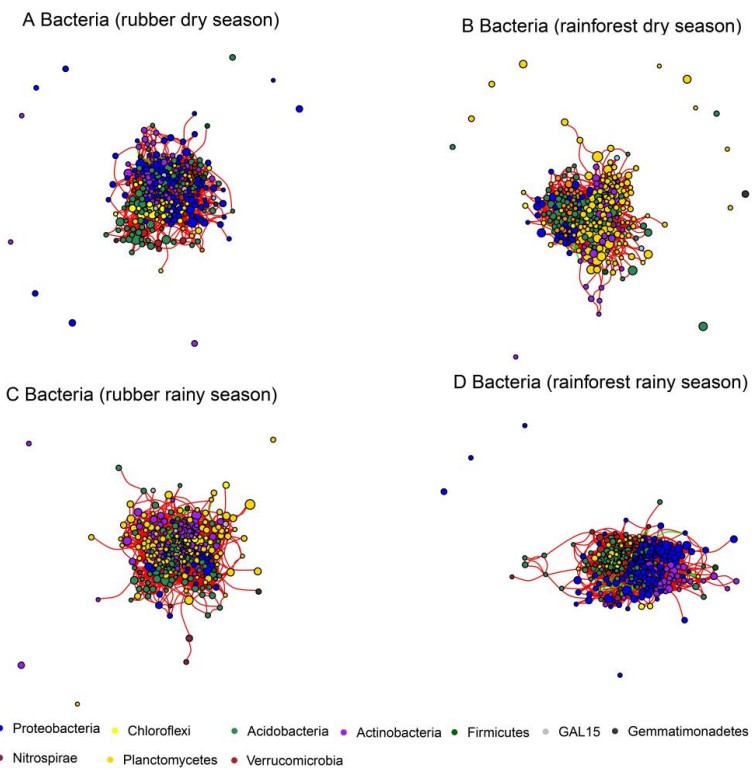


**Figure 1** Soil bacterial network structure of rubber plantations and tropical rainforest in dry and
rainy season. Red line indicates positive correlation between OTUs, and green indicates negative
correlation. Absolute value of correlation coefficient > 0.6, p < 0.05





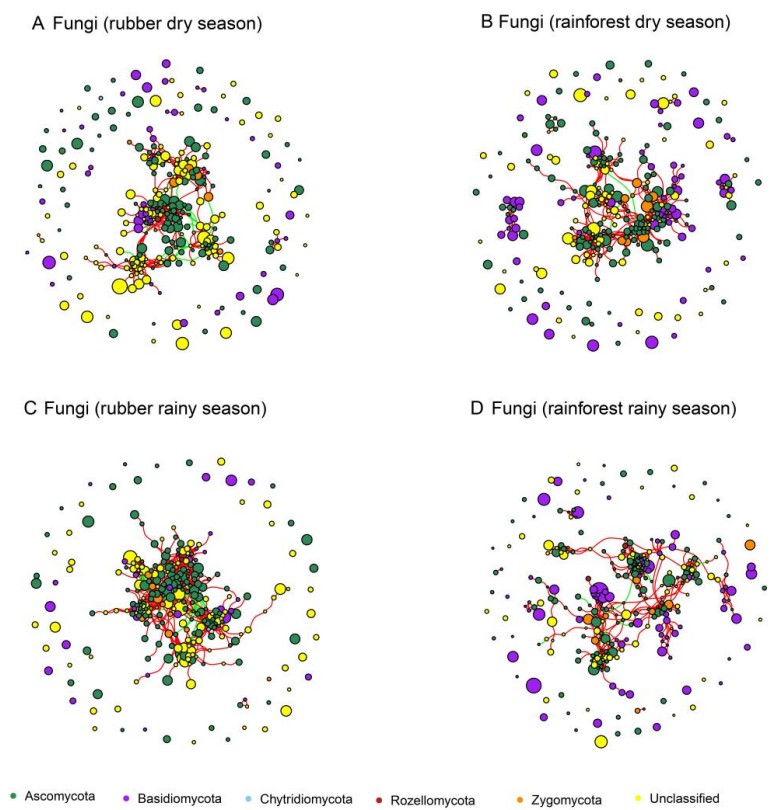


**Figure 2** Soil fungal network structure of rubber plantations and tropical rainforest in dry and
rainy season. Red line indicates positive correlation between OTUs, and green indicates negative
correlation. Absolute value of correlation coefficient > 0.6, p < 0.05





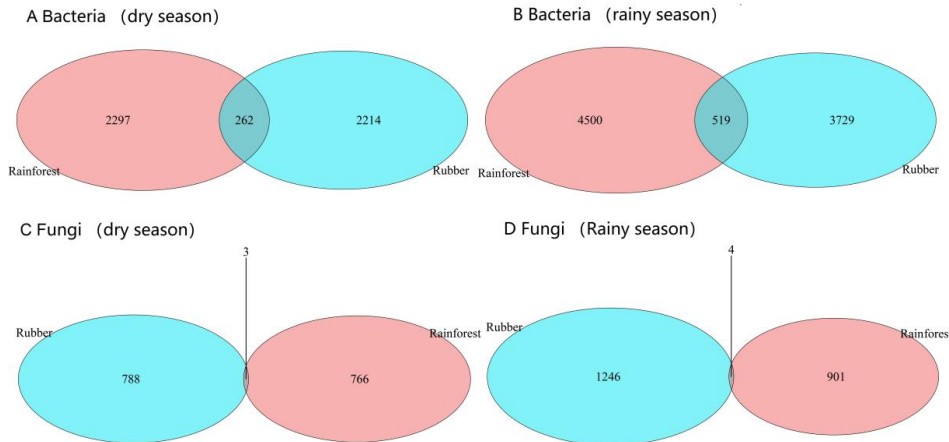


**Figure 3** Soil microbial (bacterial and fungal) network of tropical rainforest and rubber plantations
in dry season and rainy season.



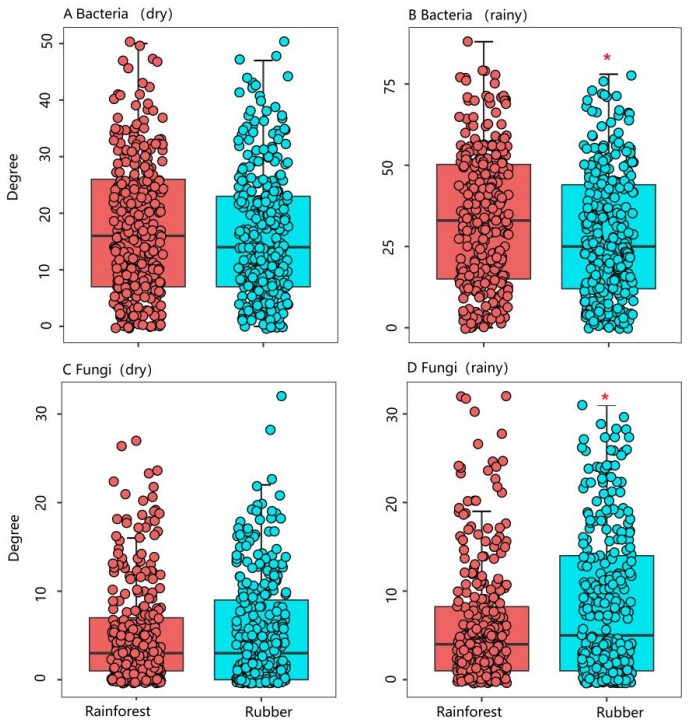


**Figure 4** Network degree betweenness of soil bacterial and fungal community of rubber
plantations (blue) and tropical rainforest (red) in dry season and rainy season.







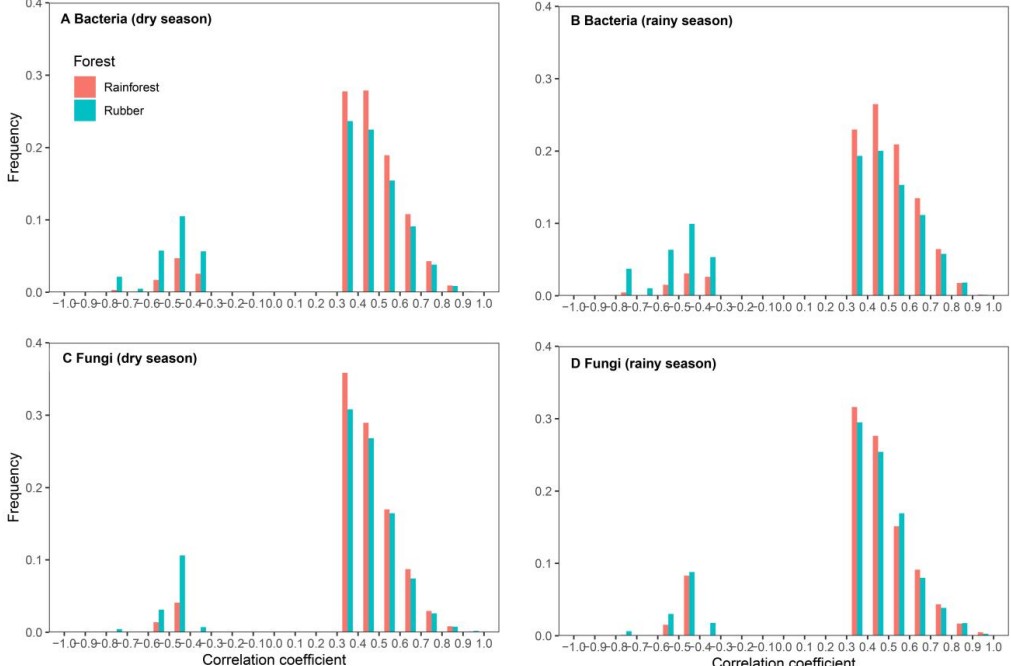


**Figure 5** Frequency distributions of correlations in bacterial (a: dry season, b: rainy season) and
fungal (c: dry season, d: rainy season) networks of rubber plantations and tropical rainforest in dry
season and rainy season. (Absolute correlation coefficient greater than 0.3, $p < 0.05$) Correlations
in rainforest networks are red, correlations in rubber plantation networks are blue.







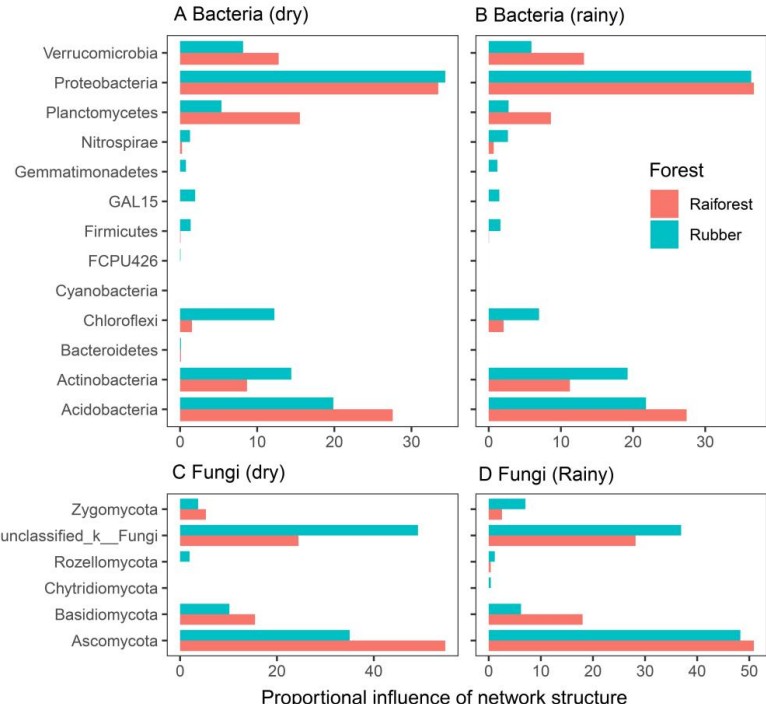


**Figure 6** Proportional influence of different phylum on bacterial and fungal network structure in both dry season and rainy. The influence was the number of degrees of nodes belonging to a particular phylum. (a: bacteria in dry season, b: bacteria in rainy season, c: fungal in dry season, d: fungal in rainy season).





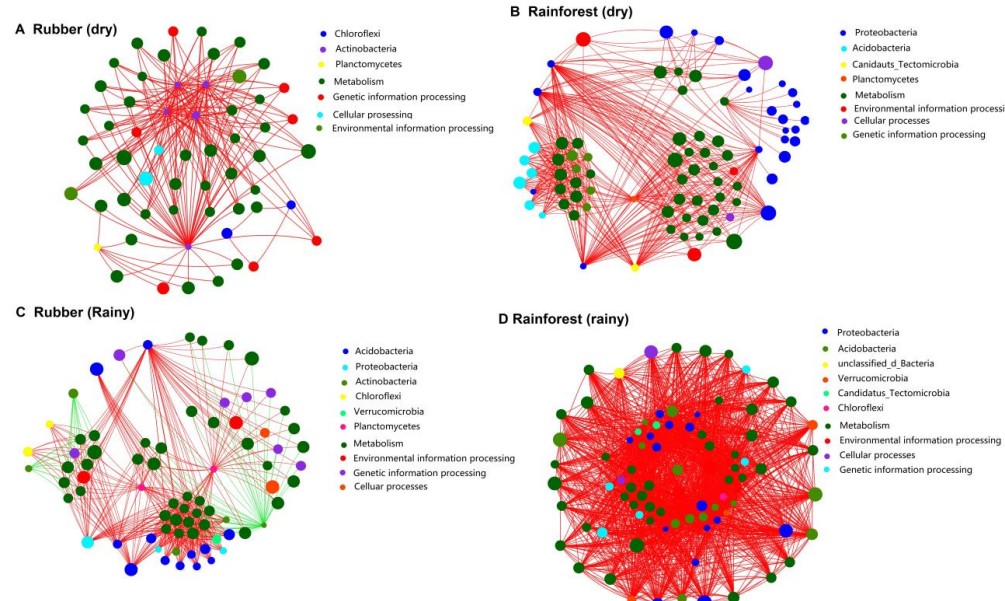


**Figure 7** Network of the top 50 abundant species (based on metagenomics data) and top 50 abundant KEGG function (pathway level 3) of rubber plantations and tropical rainforest in dry season and rainy season. (A: rubber in dry season; B: rainforest in dry season; C: rubber in rainy season; D: rainforest in rainy season) The size of the node indicates the species/function abundance. Red line indicates positive correlation between species/functions, and green indicates negative correlation. Absolute value of correlation coefficient > 0.6, p < 0.05







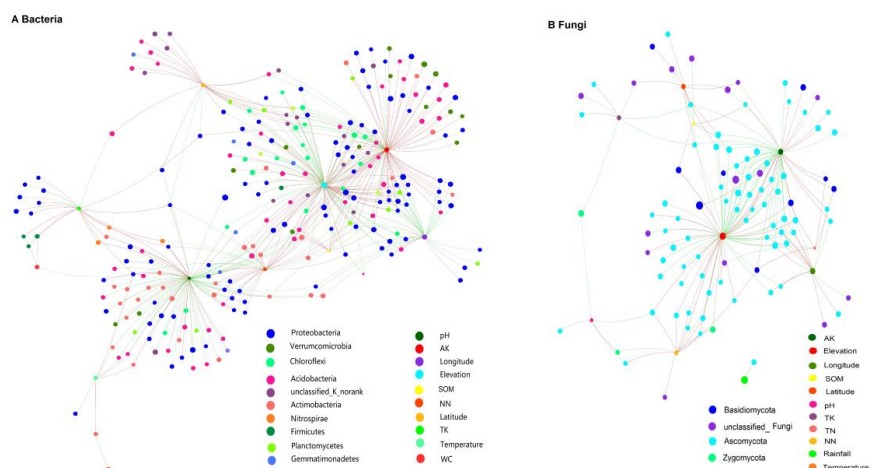


**Figure 8** Two ways correlation network of top 500 abundant bacterial (A) and fungal (B) OTU
and environmental factors. The size of the node indicates the OTU abundance. Red line indicates
positive correlation between species/functions, and green indicates negative correlation. Absolute
value of correlation coefficient > 0.5, p < 0.05.



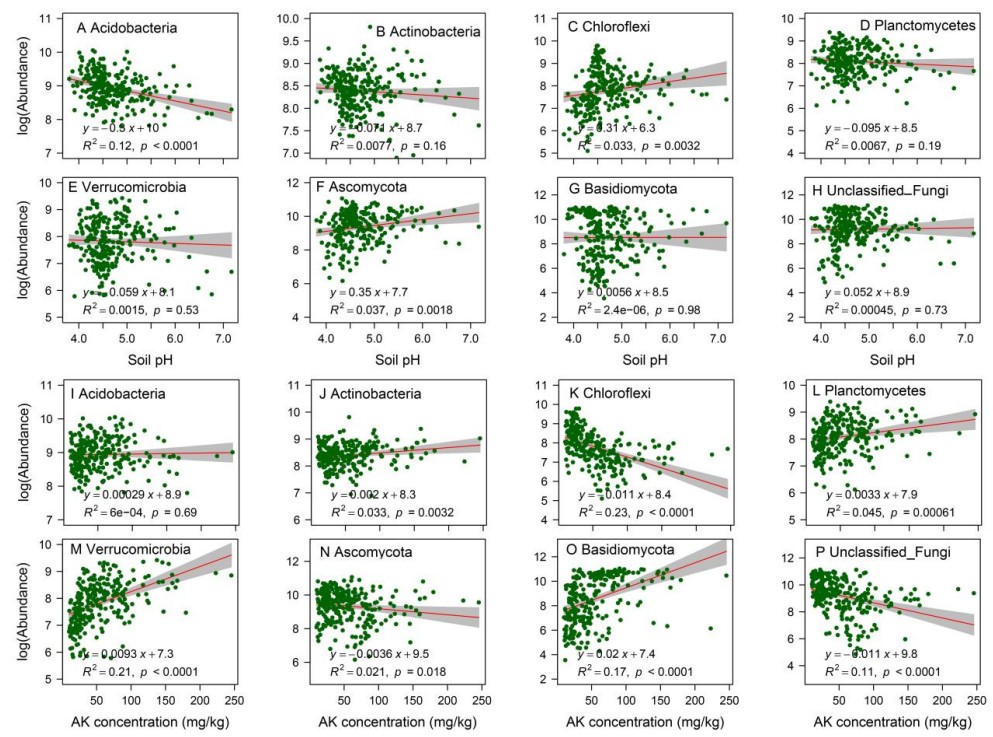



**Figure 9** Relationship between abundance of phylum (bacteria: A-E, I-M; fungi: F-H,
N-P) and soil properties (Soil pH: A-H; AK (available potassium) concentration: I-P)