# Peer review of "Network complexity of rubber plantations is lower than tropical forests for soil bacteria but not fungi"

_SOIL, 2021_

## Author Response (AR1)

**Manuscript number:** Soil-2021-98

**Title:** Network complexity of rubber plantations is lower than tropical forests for soil bacteria but not fungi.

**Journal title:** SOIL

On behalf of my co-authors, we appreciate Dr. Hogan very much for his positive and constructive comments and suggestions on our manuscript entitled "Network complexity of rubber plantations is lower than tropical forests for soil bacteria but not fungi" submitted to SOIL. We have studied the reviewer' comments carefully and made the revision according the comments of the reviewers. The following are major changes (in blue) in the revised MS and responses to the comments.

**Main comments:**

I found the introduction of network analysis lacking in explanation. The reason we write is to explain. For those who are not familiar with network analyses (and even for those who are), you should introduce key concepts – like network complexity – in an explanatory fashion. What is it? Why is it important? How does it apply in the context of the study as a tool for hypotheses testing?

Response: Tanks for the comments. I have explained the key concept of network as well as how to apply network as a tool for hypotheses testing.

Microbial network consists of two parts: nodes and edges. Nodes indicate microbes, such as OTU or species, and they can also indicate environmental variables we interested, such as soil pH. Edges (or links) indicate significant correlations between nodes. The number of links related to a node is the node's degree (Röttjers and Faust, 2018). Usually, the more links of the network has, the more complex the community is. Similarly, the higher the degree of a node (OTU or species), the more important the species is to the network structure (Berry and Widder, 2014). In recent years, microbial network analysis has been used to identify keystone taxa (Banerjee et al., 2018) and provide comprehensive insight into the microbial community structure and assembly (Fuhrman, 2019). The complexity of network structure is understood by calculating the number of edges, and the keystone species of the community are found by estimating the degree of species. The stability of the microbial community is determined by calculating the ratio of positive and negative correlation of the network

because a large proportion of positive correlation for microbial community are deemed to be unstable (Coyte et al., 2015).

Reference:

Fuhrman, J. A. Microbial community structure and its functional implications. Nature 459, 193-199, 2009

I found the hypotheses to be somewhat superficial in their scope. Simply testing for differences in the composition of fungi and bacteria among sites is a poor way to frame investigations that use soil sequencing data. Indeed, I think the study had more refined aims, and clearly articulating the specific hypotheses (in terms of fungal and bacterial guilds which may be selected for, or how compositional network complexity, etc. may change under certain soil environments and why) should be considered. Having specific, testable hypotheses not only strengthens the science but also increases the interest of readers.

Response: Very nice comments. We have revised the hypotheses as follows:

Previous study conducted in Hainan Island showed forest conversion from rainforest to rubber plantations resulted in shifts in bacterial composition from the Proteobacteria to Chloroflexi, and fungal composition from Basidiomycota to Ascomycota (Lan et al., 2020a). The above research also shows microbial (including bacteria and fungi) diversity was higher in rainforest soils than in rubber plantation (Lan et al., 2020a). Therefore we hypothesize that (1) Due to forest conversion from rainforest to rubber plantations results in changes in dominant phyla of microbes in soils, the network structure and related keystone species also changed accordingly. (2) Soil microbial (including bacterial fungi) network structure were less complex but more stable in rubber plantation than in rainforest due to high species diversity leads to complex network structure and unstable microbiome communities (3) Soil bacterial network in rubber plantation is less complex than rainforest because intensive cropping reduces the complexity of bacterial network although the richness is increased (Karimi et al., 2019).(4) Due to returning litter to the field and applying organic fertilizer in rubber plantation, soil fungal network structure in rubber plantation sites is more complex than in rainforest because organic farming showed a much more complex fungal network than conventional or no-tillage farm systems (Banerjee et al., 2019).

Reference:

Karimi, B., Dequiedt, S., Terrat, S., Jolivet, C., Arrouays, D., Wincker, P., Cruaud, C., Bispo, A., Prévost-Bouré, N.C., Ranjard, L., Biogeography of soil bacterial networks along a gradient of cropping intensity. Scientific Reports, 9(1) 3812, 2019.

Similarly, the discussion of the potential abiotic soil drivers and of these differences should be expounded upon, in my opinion (see comments below).

Response: Thanks for the comments. We have revised the discussion of the potential abiotic soil drivers and these differences.

The methods used for (abiotic) soil analyses need to be included (see comments below).

Response: The methods used for abiotic soil analyses were added in the revised manuscript.

Some justification must be given here for "only focusing in the top 300 most abundance OTUs" (L170). I can understand why this was done – because it simplifies analyses, however, there is much evidence that suggests that abundances of OTU reads are not indicative of naturally occurring abundance of fungi or bacteria, because for example, primer specificity, amplification preferences during PCR and host of other sources of selection variability/error that can occur in these types of dataset. I think those should at least be acknowledged. Has the analysis been explored using different subsets of the data (say the 300 most common OTUs, or the entire dataset)? did results differ? This comment also applies to other subsets of the data used in the statistical analyses (e.g., L189: "the top 50 most abundant species" etc.)

Response: Very good comments. To make the analyses simple, we only use the top 300 most abundant OTUs to analyze the network structure, and these OTUs are approximately equal to OTUs with relative abundance greater than 0.05% (Jiao et al., 2016).

Reference:

Jiao, S., Liu, Z.S., Lin, Y.B., Yang, J., Chen, W.M., Wei, G.H. Bacterial communities in oil contaminated soils: biogeography and co- occurrence patterns. Soil Biol. Biochem. 98:64-73, 2016

To make sure the results is correct, it is necessary to use different data to test results.

We have used different data (top 300 most abundant OTU and top 500 OTU) to perform the network analysis before we submitted this paper. The results were very similar. To simply the analysis, we only used the top 300 OTU for network in our study. The follows are the main results of the top 500 most abundant OTU.

For the top 500 most abundant OTU, The results showed that soil bacterial community network was more complex in rainforest (6103 edges in in dry season, 12120 edges in rainy season) than in rubber plantations (5963 edges in in dry season,, 11807 edges in rainy season), while fungal community network was more complex in rubber plantation (2297 edges in in dry season, 3664 edges in rainy season) than in rainforest (1963 edges in in dry season, 2407edges in rainy season). In addition, soil microbial network was more complex in rainy season than in dry season.

Similarly, the top 50 species were used to make network analysis simple and the network figure easily to read (please see figure 7d). If more species were used, the network would not be easily read.

Line comments / technical corrections:

**ABSTRACT**

L28: You might define/ explain briefly what network structure means in the context of the main result, here.

Response: We have added a sentence in abstract as follows:

Microbial network is viewed as a critical indicator of soil health and quality, and consists of two parts: nodes and edges.

L37: I found this sentence to seemingly jump out of the abstract without previous reference. This does not seem like a logically flowing conclusion from the previous eight sentences of the abstract. Nothing about soils was mentioned. If the main conclusion about how available K and total N drive the community/ network structure, you should mention how they vary across rubber vs. natural forests.

Response: We have revised the abstract.

Further analysis shows soil pH, potassium (AK), total nitrogen (TN) had more links with species of some phyla. Inclusion, forest conversion results in an increase in soil

pH, and a decrease in AK and TN, and these changes as well as seasonal variations had a great impact on soil microbial composition, network structure and function.

**INTRODUCTION**

L54: Needs space after comma.

Response: Done.

L55: Hainan Island, China

Response: Done.

L56: at the northern edge of Asia's rainforest distribution.

Response: Thanks. Done.

L59: You might want to define soil microbiome, just to be clear about what you mean.

Response: Soil microbiome is the generic term of massive microorganisms and complex soil environment. Therefore, we changed the sentence as follows:

Soil microbiome is the generic term of massive microorganisms and complex soil environment and it is highly diverse and comprises up to one quarter of Earth's diversity.

L67-73: See studies by Song HK et al. FEMS https://doi.org/10.1093/femsec/fiz092 and Ma H et al Forests 2019 https://doi.org/10.3390/f10110978

Response: Thanks for the literatures. We have read the literatures and added these reference in the text.

Compared to Eucalyptus plantations, rubber plantations have higher diversity of both bacteria and fungi (Ma et al., 2019)

Song et al. (2019) reported that tropical forest conversion to rubber plantation results in reduced fungal microbial community network complexity, while there are few studies on the impact of forest conversion on soil bacterial community network structure and the drivers leading to the changes of network structure.

Song H., Singh, D., Tomlinson, K.W., Yang, X.D., Ogwu, M.C., Slik, J. W. F., Adams, J.M. Tropical forest conversion to rubber plantation in southwest China

results in lower fungal beta diversity and reduced network complexity, FEMS Microbiology Ecology, 95, 7, fiz092,2019

Ma, H., Zou, W., Yang, J., Hogan, J. A., Chen, J. Dominant tree species shape soil microbial community via regulating assembly processes in planted subtropical forests. Forests, 10(11), 978, 2019

L76: "may represent system behavior" – what does this phrase mean? You might give an example or further explain/define this.

Response: We have rewritten the most part of the introduction. This sentence is unnecessary, so we deleted it

L80: "standard suite of analytical approaches" – such as?

Response: We have rewritten the most part of the introduction. This sentence is unnecessary, so we deleted it

L93: "alters microbial community composition" This is a vague hypothesis. One major criticism of these types of sequencing papers is that they test the hypothesis of difference among sites. Indeed with thousand of taxa/ OTUs, you will likely find differences. This is not a very ecologically meaningful or interesting hypothesis. Surely, with all the work that has been done on how rubber plantations affect soils, you had a more refined hypothesis? What bacterial or fungal taxa/ guilds did you think might be driving differences?

Response: We have revised this part corroding the comments, please see above.

L98: "clarify the drivers" This is the more-novel / more-important part of the paper in my opinion. Identifying the drivers of why soil fungal and bacterial communities are affected by forest conversion to rubber plantation and plantation management has implications for real-world ecology. However, there is no discussion of the potential drivers in the introduction. Soil chemistry, moisture, etc are key considerations, which is influenced by litter quality and alterations to biogeochemical cycling as a result of the conversion of forest to rubber plantations. A brief overview of this might be helpful in setting up this hypothesis better.

Response: Thanks for the comments. We have added the followings according the comments.

While the exact drivers of microbial network structure still remain unknown. Previous study showed that soil nutrients, such as soil phosphorus content, and soil pH, are the main drivers for the network structure of microbial community (Banerjee et al., 2019). Seasonal variation also affects the network structure by changing the keystone species of the community because a keystone species might be present only in a specific season or time period (Banerjee et al., 2018).

**METHODS**

L88-120: Ling translates to mountain from Chinese to English, but you should still use the full Chinese names of the places (in English). Bangwangling, Jianfengling etc. These are the names of the places.

Response: Thanks for the comments. Revised.

L121: On your map (Figure S1) you should label the sites, so people know, where each of the named study sites is. For example, where Bawangling vs. Jianfengling.

Response: Thanks for the comments. Revised.

L122: What was the depth of the soil sampling? What type of soil instrument was used (Give details on diameter etc). How was sterility maintained between soil sample collections? These are important missing details.

Response: The top soil (0-20 cm).

Before soil sampling, we sterilized the soil drill with 75% alcohol. After the removal of the litter layer, by using a 5-cm diameter steel drill, top soil (0 to 20 cm) was collected, then homogenized and passed through a 2-mm mesh sieve.

L126: were soils sieved?

Response: Yes.

L130: being as this is a soils discussion journal, you should describe briefly what methods and instruments were used to measure soil nutrients. It may be okay to refer to the Lan et al 2020 reference for some of the finer details, but you should give enough information to not leave readers guessing. No details are given, which is suspect.

Response: Thanks for the comments.

Soil water content (%) was measured gravimetrically. Soil pH was measured in a soil/water suspension (1: 2.5, w/w) using a pH meter. Soil total nitrogen (TN) was determined using a micro-Kjeldahl digestion followed by steam distillation. Total phosphorus (TP) and total potassium (TK) were measured following digestion with NaOH. Nitrate nitrogen (NN) and ammonium nitrogen (AN) were determined by steam distillation and indophenol-blue colorimetry, respectively. Soil samples were extracted with NaHCO3 and the extract was then used to measure available soil phosphorus (AP) via molybdate-blue colorimetry. To measure soil available potassium (AK), soil was extracted with ammoniumacetate and then the extract was loaded onto an atomic absorption spectrometer with ascorbic acid as a reductant (Chen et al., 2019).

L132-168: sequencing methods read well & were easily followed/understood.

Response: Thanks for the positive comments.

L183: you might define what a "keystone OTU" is (briefly and generally), before explaining how they were selected.

Response: We have revised the sentence as follows:

Keystone OTU are known to be important for ecosystem structure and function and were selected on the basis of high network degree, high closeness centrality, and low betweenness centrality as defined by Berry and Widder (2014)

**RESULTS**

The results generally seemed solid and well presented. I like the use of the correlation analysis – relating positively and negatively correlated OTUs to one another within the framework of the network analysis.

Response: Thanks for the positive comments.

I think certain figure legends could be elaborated. For example, from the figure legend for Fig. 3, it is unclear what is being shown in the graphic.

Response: We have revised figure legends of Figure 3 as follows:

Figure 3 Number of shared and unique edges of soil bacterial and fungal networks in rubber plantations and tropical rainforests in the dry and rainy season. The number where the two circles cross is number of shared edges. Numbers in red circle presents

the unique edges in rainforest, while in blue circle present the unique edges in rubber plantations.

L249 (and elsewhere) Chloroflexi (with an i at the end).

Response: Sorry for the errors. Revised.

Also, throughout the results, species are referred to as "members" of certain taxonomic groups. I think could use the word species, although this is a matter of personal preference. Members sounded a bit odd to me, personally.

Response: Thanks for the comments. Revised.

**DISCUSSION**

The two papers I have linked to above (Song et al., and Ma et al.) should be incorporated into the discussion (e.g., L318 and elsewhere).

Response: We have added these two literatures in the discussion as follows.

Our results were not consistent with a study conducted in Xishuangbanna (Song et al., 2019) which showed that tropical forest conversion reduced fungal network complex, but were consistent with other previous observations which found that fungal community networks were better organized disturbed forest compared to primary forest (Chen et al., 2019).

The discussion seemed adequate for the most part. I found it a bit superficial. The authors might try to weigh in more on the actual ecological significance of some of the changes they found. What does it mean for soil biogeochemical cycling or ecosystem functioning? For example, does losing some Actinobacteria from soils from natural forests to rubber plantations actually matter? There is a lot of functional redundancy among soil bacteria and fungi, especially in the tropics, so what are the potential actual consequences for such changes in the soil microbiome? I know this may seem speculator, but it's interesting to discuss this, in my opinion, even if briefly. Also, what future research directions might be informed by the findings of this study.

Response: Thanks for the comments.

**4.5 Possible impact of forest conversion on microbial community function**

Forest conversion results in a decrease in abundance of Proteobacteria and increase of Actinobacteria. Most species of Proteobacteria was positively correlated with metabolic function, while most of Actinobacteria was negatively associated with metabolic function (Figure 8). Therefore, the changes in the abundance of these two phyla results in a reduction of microbial community function after forest conversion. Due to metabolic function of a specific species usually affected by environmental conditions (Louca et al., 2018), some species are not correlated with any function in the dry season, but correlated with metabolic function in the rainy season, indicating there some microorganisms do not participate in the metabolic process in dry season, especially for the rubber plantations. In conclusion, forest conversion as well as seasonal variation had a great impact on soil microbial community functions.

Reference:

Louca, S., Polz, M.F., Mazel, F., Albright, M.B.N., Huber, J.A., O'Connor, M.I., Ackermann, M., Hahn, A.S., Srivastava, D.S., Crowe, S.A., Doebeli, M., Parfrey, L.W. Function and functional redundancy in microbial systems. Nature Ecology & Evolution. 2, 936-943, 2018

**Manuscript number:** Soil-2021-98

**Title:** Network complexity of rubber plantations is lower than tropical forests for soil bacteria but not fungi.

**Journal title:** SOIL

On behalf of my co-authors, we appreciate reviewer 2 very much for his positive and constructive comments and suggestions on our manuscript entitled "Network complexity of rubber plantations is lower than tropical forests for soil bacteria but not fungi" submitted to SOIL. We have studied the reviewer' comments carefully and made the revision according the comments of the reviewers. The following are major changes (in blue) in the revised MS and responses to the comments.

**Main comments:**

This study did a comprehensive investigation on soil bacterial and fungal networks in response to tropical forest conversion, by comparing the network degree within microbial community and between microbiomes and environments under protected rainforests with those under rubber plantations. The author demonstrated a simpler bacterial network while a more complex fungal network in the rubber plantations, mainly through comparing the network degrees. The idea is novel, the method is reasonable, and the main results can advance the understanding of soil microbial shifts caused by forest conversion in tropical areas and help with the management strategies in terms of soil system. Nevertheless, I have some minor issues on the manuscript organization that should the author concern before accepted by SOIL.

1) Too much description of tropical biodiversity (both above and below ground communities) in the introduction make it difficult to concentrate on the hypotheses.

Response: Thanks for the comments. We have rewritten most of the introduction.

2) Some definitions and expressions in methods need further clarification, such as sampling interval, shared edges, keystone taxa, etc.

Response: Thanks for the comments.

Soil sampling was performed twice in 2018, once in January (dry season) and once in July (rainy season).

The number of shared edge and unique edge as well as keystone OTU were calculated to evaluate whether the network structure has changed. Shared network edge is the link (edge) between species A and species B not only exists in rubber plantation network, but also in rainforest network. Unique edge is the link only existing in rubber plantation or rainforest. Keystone OTU are known to be important for ecosystem structure and function and were selected on the basis of high network degree, high closeness centrality, and low betweenness centrality as defined by Berry and Widder (2014)The bacterial-fungal community network analysis were performed to investigate soil microbial network complexity.

Moreover, the bacterial-fungal interkingdom network analysis is proposed to investigate soil microbial network complexity.

Response: We have performed the bacterial-fungal community network analysis in the revised manuscript and the results were as follows:

The bacterial-fungal community network were more complex in rubber plantation (4284 edges in dry season, 7257 edges in rainy season) than in rainforest (3650 edges in dry season, 6507 edges in rainy season), and more complex in rainy season than in dry season. The results further revealed that rubber plantations (844 edges in dry season, 1744 edges in rainy season) have more negative links than rainforest (149 edges in dry season, 489 edges in rainy season) indicating network of rubber plantation was more stable than rainforest.

3) As the author investigates the connections of microbial communities with soil nutrients content and functional groups, further explorations about the potential effects on ecosystem functioning caused soil microbial network shifts might be important.

Response: Thanks for the comments. We have discussed the potential effects of network shifts on ecosystem functions.

Most species of Proteobacteria was positively correlated with metabolic function (Figure 7). Therefore, the reduced complexity of soil bacterial network structure in rubber plantation was mainly due to the reduction of the proportion of Proteobacteria. Due to metabolic function of a specific species usually affected by environmental conditions (Louca et al., 2018), some species are not correlated with any function in the dry season, but correlated with metabolic function in the rainy season, indicating

there is a lot of functional redundancy in microbial community in dry season, especially for the rubber plantations.

4) English Grammars and some word expressions need to be improved.

Response: Thanks for the comments. We have read the manuscript carefully and eliminated many small errors.

**Detailed comments/technical corrections:**

L21: rainforest should be rainforests.

Response: Done.

L22-23: we used the data from Illumina sequencing and metagenome shotgun sequencing….

Response: Done.

L25: please clarify the "shared network edges".

Response: Here we defined shared network edge is that the link (edge) between species A and species B not only exists in rubber plantation network, but also in rainforest network.

L32: in rubber plantations…; please remove "higher" before links.

Response: Done.

L33: forest conversion increased fungal network complexity.

Response: Done

L34-35: maybe it is more clear as "The keystone taxa in bacterial networks shifted from Acidobacteria in rainforests to Actinobacteria in rubber plantations".

Response: thanks for the comments. We have rewritten abstract.

L37-39: it is not clear for the relationships between soil properties and microbial network structure, Please rewritten the conclusion sentence.

Response: Thanks for the comments. We have rewritten abstract.

Further analysis shows soil pH, potassium (AK), total nitrogen (TN) had more links with species of some phyla. Inclusion, forest conversion results in an increase in soil

pH, and a decrease in AK and TN, and these changes as well as seasonal variations had a great impact on soil microbial composition, network structure and function.

Please add some values when describe the changes in networks.

Response: Done

L93-94: remove"[3]", did you investigate soil microbial activity?

Response: Sorry for the errors. We did not investigate soil microbial activity.

L98: Drivers and mechanisms: do you mean the soil properties or relating soil processes? Please clarify.

Response: By testing these hypotheses, we want to clarify the drivers and mechanisms of microbial community assembly that link forest conversion to differences in soil microbial network structure.

L112: Please move the sentence "Rainfall is abundant, ranging from 1000 mm to 2600 mm yearly, with an average annual precipitation of 1639 mm." to L109.

Response: Thanks for the comments. Done.

L115: When the rubber plantations have been established and what are the total areas? Need general information of the forest conversion.

Response: Thanks for the comments. We have added some information in method part.

There are about 530, 000 hectares of rubber plantations in Hainan Island. We selected 25-30 years old rubber trees (i.e., mature rubber plantations) as our study objectives.

L118-120: What are the criteria when selecting these sampling sites?

The tropical rainforests in Hainan are mainly distributed in Bangwangling, Diaoloshan, Wuzhishan, Yinggeling and Jianfengling. Therefore, we selected a tropical rainforests in each of these sites as our study objectives. Five rubber plantations were selected in Wanning, Ledong, Danzhou, Haikou, Qiongzhong, and these five rubber plantation locate in the east, south, west, north and middle of Hainan, respectively.

L122: Which soil layer?

Response: After the removal of the litter layer, by using a 5-cm diameter steel drill, top soil (0 to 20 cm) was collected, then homogenized and passed through a 2-mm mesh sieve.

L123: What means sampling interval?

Response: Sorry for the errors. Deleted "per sampling interval"

L127: Soil water content; please specify the sample store conditions;

Response: Thanks.

The other was stored in ultra-low temperature (- 80 ℃) refrigerator for later DNA extraction.

L138: archaeal community was not included in the following analysis;

Response: Sorry for the errors. Deleted "archaeal".

L139: The sequence data should be deposited in an online dataset, such as NCBI;

Response: Thanks for the comment.

The raw reads were deposited into the NCBI Sequence Read Archive (SRA) database (Accession Number: SRP108394, SRP278296, SRP278319). In the revised manuscript, we put this sentence in the method part.

L183, L196-199: The connectors, module hubs and network hubs have been commonly identified as keystone taxa in network in many studies, what are the differences between these network groups and the keystone taxa that you identified in 183?

Response: Thanks for the comments. We agree that module hubs and network hubs have been commonly identified as keystone taxa in network. However, in our study, there were no module hubs and network hubs (please see figure S3 and figure S4). We adopted another standard, that is high degree, high closeness centrality and low betweenness centrality.

L228: What means "more correlations"? Pease clarify.

Response: deleted "more".

237-239: Maybe the statistical comparisons of network parameters should be applied to obtain this result.

Response: Thanks for the comments.

The whole results section is wordy, to simplify by concentrating on the main results, we have deleted this sentence.

L284-288: Need statistical values or network parameters when comparing the network complexity.

Response: Thanks for the comments. The results showed that soil microbial network structure in at rainforests sites (460 edges in dry season, 1750 in rainy season) was more complex than rubber plantations (223 in dry season, 451 in rainy season) indicating that more links between microbes and function were observed in soils of rainforest..

The whole results section is wordy, Please simplify by concentrating on the main results.

Response: This part has been simplified.

L392-394: Did the rubber plantation received any fertilizer?

Response: Yes. We added this information in method. Management practices, such as latex harvest and the application of fertilizers, are used in rubber plantations. Usually, compound fertilizer (1-1.5 kg per tree) and organic fertilizers (20-25 kg per tree) were applied once or twice a year.

L420: impact on

Response: Done.

L421: What kind of implications for ecosystem functions? Could you please be more specifically?

Response: Our study demonstrates the impact of forest conversion for soil network structure, which has important implications for ecosystem functions, such as metabolic function, of soil ecosystems in tropical regions.

Figure 8: Maybe it is better to use different symbol to display environmental variables.

Response: We have redone Figure 8.

---

## Author Response (AR2)

**Manuscript number:** Soil-2021-98

**Title:** Network complexity of rubber plantations is lower than tropical forests for soil bacteria but not fungi.

**Journal title:** SOIL

**Comments to the author**:

The revised manuscript has now been reviewed by one of the original reviewers. The reviewer found the MS to be improved, and scientifically sound, but has concerns about the quality of writing. I agree that the writing is currently not clear or grammatically correct in many areas of the manuscript and needs improving for an international audience before it can be accepted for publication. While the decision is minor revision, the manuscript will not be accepted if the grammatical issues are not corrected.

Examples:

Line 76 "Microbial network consists of "should be "microbial networks consist of ".

Line79 "environmental variables we interested" is not correct should be "environmental variables of interest".

These are just two examples, such issues are pervasive throughout the manuscript.

Response: On behalf of my co-authors, we appreciate Dr Ember Morrissey and Dr. Hogan very much for his positive and constructive comments and suggestions on our manuscript entitled "Network complexity of rubber plantations is lower than tropical forests for soil bacteria but not fungi" submitted to SOIL. We have asked a native English speaker (Tim Treuer) to revise it for one more time. We believe that the writing of this paper has been greatly improved.